# HIV, HTLV, and Hepatitis B and C Infection in Blood Donors in Bahia, Brazil from 2008 to 2017

**DOI:** 10.3390/v14112323

**Published:** 2022-10-22

**Authors:** Estela Luz, Marinho Marques, Eduardo Martins Netto, Luisa Meireles Campos, Sávio Amaral, Iraildes Santana, Eduardo Luz Marques, Carlos Brites

**Affiliations:** 1Programa de Pós-Graduação em Medicina e Saúde, Universidade Federal da Bahia, Salvador 40110-060, BA, Brazil; 2Laboratório de Pesquisa em Infectologia, Hospital Universitário Professor Edgard Santos, Salvador 40110-060, BA, Brazil; 3Fundação Bahiana de Infectologia, Salvador 40110-160, BA, Brazil; 4Fundação de Hematologia e Hemoterapia da Bahia, Salvador 40286-240, BA, Brazil; 5Departamento de Ciências da Vida, Universidade do Estado da Bahia, Salvador 41180-045, BA, Brazil; 6Escola Bahiana de Medicina e Saúde Pública, Salvador 40290-000, BA, Brazil

**Keywords:** HIV, HTLV, HCV, HBV, blood donors, transfusion-transmitted infections, prevalence, incidence

## Abstract

Although blood transfusion is an important therapeutic resource, transfusion-transmitted infections (TTIs) are still a cause for concern. Measures to mitigate this risk involve improvement of donor screening criteria and improvements in laboratory tests, especially the use of nucleic acid test (NAT). In this retrospective study we evaluated HIV, HTLV, HCV and HBV infection rates in blood donors of the Hematology and Hemotherapy Foundation of Bahia (Hemoba), Brazil, through serological and NAT results and the characteristics of donors. From February/2008 to December/2017, 777,446 blood donations were made. Most donors were male, aged 25–44 years, black and mixed race, and single or divorced. The density-type incidence (DTI; per 100,000) for each virus was 91.1 for HBV; 66.5 for HCV; 54.3 for HIV; and 33.9 for HTLV, with a decreasing trend observed over the period studied, except in the last biennium. NAT detected only 1 donor in immunological window for HIV (0.46/100,000 donations) and 3 donors in immunological window for HBV (1.8/100,000 donations). Serological positivity for all viruses studied was higher in the metropolitan region of Salvador, the state capital. Conclusion: DTI rates show a decreasing trend over the years studied, with a predominance of HBV infection. NAT allowed the detection of donors in immunological window periods, having an important role in improving transfusion safety.

## 1. Introduction

Annually more than 120 million blood units are donated worldwide, and in Brazil, there are more than 3.5 million voluntary blood donations per year [1]. It is widely known that blood transfusion plays a very important role in healthcare [2]. However, the safety of blood products remains a great concern due to the risk of transfusion-transmitted infections (TTIs) [3]. The World Health Organization (WHO) establishes that every blood donation should be screened for Human Immunodeficiency Virus (HIV), Hepatitis B Virus (HBV), Hepatitis C Virus (HCV) and Syphilis [1]. In Brazil, in addition to these infections, a screening is also mandatory for Human T Lymphotropic Virus (HTLV) and Trypanosoma cruzi as well as, for malaria, in endemic areas. In addition, a clinical screening is performed to evaluate clinical history and lifestyle of the blood donor candidate to decrease the risks of TTIs, by preventing donation from those candidates at-risk for such infections [4,5,6].

According to the Brazilian Institute of Geography and Statistics (IBGE) [7], the state of Bahia (located in the Northeast region of Brazil) has an estimated population of 14,985,284 people distributed in 417 municipalities. According to data released by the Ministry of Health, the state of Bahia had 193,053 blood donations in 2016. The Hematology and Hemotherapy Foundation of Bahia (Hemoba) is the official government hemotherapy service in Bahia and is responsible to coordinate the National Blood Policy within the state. In addition to that it has 29 units in different cities (attending all regions of the state) representing 53.6% of all the blood collected in the state during 2016. Private hemotherapy services is a complementary to this net [8].

With the implementation of new serological and molecular tests with higher sensitivity and specificity, TTIs have become extremely rare, increasing the safety of the recipient [9,10,11]. The WHO defines window period as the time between the infection and the moment of a positive result in the tests [4]. To reduce this time as much as possible, allowing fewer TTIs in the window period, nucleic acid test (NAT) was incorporated in the screening process [11]. The use of NAT for HIV and HCV screening became mandatory in Brazil in 2014 and for HBV in 2016 [12,13].

The rates of reactive and non-reactive serological tests reflect variables of great importance to ensure the quality of blood in transfusions. Serological screening of blood donors has been used in several countries to improve public health programs. Populational studies to verify the rates of such infections are surrounded by ethical, operational, and financial difficulties [5,14]. In the present study, we analyzed the incidence and prevalence of HIV, HTLV, hepatitis B and C infection in blood donors from Hemoba in each region of Bahia, from 2008 to 2017.

## 2. Materials and Methods

### 2.1. Study Design and Population

This is a descriptive retrospective study carried out with data from February 2008 to December 2017. The rates of HIV, HTLV, HCV and HBV infections among blood donors were analyzed with pre-established criteria that specified age (16 to 69 years old), and weight (≥50 kg). All donors answered a standard questionnaire assessing current health status and possible risky behavior for blood-borne infections during the months prior to donation (time-variable depending on the issue involved). Then, only donors without risky behavior for blood-borne infections and in good health were able to donate blood.

Data were extracted from the Hemovida software, used by Hemoba, provided by the Ministry of Health. This software aims to control and monitor blood donations at Hemoba. Serological and NAT tests were performed at Hemoba laboratory located in Salvador, capital of the state of Bahia. Donors were categorized into first time donors and repeat donors. First time donors were defined as donors who had their first blood donation during the study period, while repeat donors were those who had more than one blood donation, with at least one blood donation occurring during the study period. For the calculation of DTI, were also included those negative donors with a repeat donation during the study period. A seroconverting donor (incident) was defined as a donor who made a seropositive donation during the period of the study (2008–2017) and had previously made a seronegative donation.

### 2.2. Laboratory Tests

During the study period, all donations were tested for anti-HIV-1/2, anti-HTLV-1/2, HBsAg, anti-HCV, and anti-HBc by Enzyme immunoassay and/or Chemiluminescence (Table 1). The NAT, in minipools of 06 samples, was introduced in August 2013 for the entire state of Bahia for HCV-RNA and HIV-RNA and in February 2015 for HBV DNA at the Coordinating Blood Center (Hemoba, Salvador) and in March 2015 for the entire state of Bahia. The results of the screening tests were reviewed individually for each donor, including additional testing (like western blot), according to the flowchart defined by the Ministry of Health [4]. 

Briefly, blood samples obtained from each donation are tested as defined by the Ministry of Health. Before the implementation of NAT, samples were tested in high sensitivity kits (EIA and/or chemiluminescence) to detect antigens and/or antibodies against the viruses in question, with 2 tests for HIV and HBV and 1 test for HCV and HTLV. In case of a positive result, the samples are repeated in duplicate in the same sample and an additional examination, using another methodology different from the previous ones, must be performed for confirmation. After the implementation of NAT for HIV and HBV, one of the initial tests was replaced by NAT, and for HCV, NAT was added to the serological test. Testing for HTLV remained the same, given that NAT for HTLV is not yet mandatory in Brazil.

The HBV donations were considered positive if HBsAg and/or Anti HBc and/or NAT reagent results in any combination.

### 2.3. Statistical Analysis

A database in Excel^®^ format was provided by Hemoba, and these data were later revised and recoded to allow statistical analyzes using SPSS version 24.0. The Chi square test (χ^2^) was used to compare the proportions, and a significance level of 5% (*p* < 0.05) and a confidence interval of 95% was adopted in all tests. Quantitative variables were expressed with absolute numbers and frequency. 

Demographic data were extracted from the donation records of each donor, evaluating variables such as: sex, age, race, education, marital status, and place of residence.

All analysis were performed using the number of truly positive tests. False-positive tests were considered negative after repeating the tests, according to the flowchart determined by the Ministry of Health [4].

Georeferenced information was also evaluated using the QGIZ software to identify groups with positive results for HIV, HTLV, HCV and HBV infections according to the municipality of residence, based on IBGE data (2020) [7].

### 2.4. Incidence and Prevalence Calculations

#### 2.4.1. Density-Type Incidence (DTI)

The DTI was calculated by the ratio between the number of donors who had a positive result in that biennium by the variable person-years, calculated by the time between the patient’s entry into the donation system (first donation) and their exit (last donation—regardless of their serological status) in days, considering their contribution per biennium, divided by 365 (number of days in 1 year). The result was multiplied by 100,000 and the CI was calculated by normal approximation to the Poisson distribution.

#### 2.4.2. Prevalence

The prevalence of the viruses studied was calculated using the number of first-time donors or repeat donors who tested positive during the study period divided by the total number of donations in that year and expressed per 100. The prevalence was calculated with one-year intervals for the period from 2008 to 2017. For each seroconversion episode, donor prevalence was calculated by demographic variables (gender, age, race/ethnicity, education, marital status, type of donation and area of residence) and globally. Demographic data were extracted from each donor’s donation records. Cumulative prevalence is the positive donors divided by the total donors in the period per 100 donations.

### 2.5. NAT Yield

“NAT Yield” is defined as a positive sample for NAT but negative in the serological tests. This occurs mainly due to acute infections in which the donor is still in the “window period” with detectable viremia, but still without detection of antibodies.

### 2.6. Ethical Aspects

The study was conducted according to the Helsinki’s Declaration and was approved by the Research Ethics Committee of the Maternidade Climério de Oliveira—Federal University of Bahia (protocol number: 2,597,057; 13 April 2018; CAAE: 86944718.0.0000.5543), after obtaining Hemoba’s consent. As it was a study using a database, Informed Consent Form (ICF) was not required, because no personal identifiable information was used, providing total anonymity for the donors.

## 3. Results

### 3.1. Sociodemographic Characteristics of Donors

During the period of the study, 777,446 blood donations were received from 470,964 donors (ranging from 1–102 donations/donor, including whole blood and/or donation by apheresis). Blood samples were obtained from all donations and serological tests were performed according to current legislation [4]. Most donors were male (62.8%), aged 25–44 years (61.5%), black/racially mixed (60.1%), with 8–11 years of schooling (55.8%), single and/or divorced (62.9%), who did more than one donation (53.2%) and living in inland cities of Bahia (54.2%) (Table 2). Global serological positivity was significantly higher in male individuals, aged between 25–44 years, black (black or racial mixed), with 8–11 years of schooling, living without a partner, in first-time donors and residents in inland cities of Bahia (Table 2).

### 3.2. Prevalence

As shown in Table 3, the number of donations has gradually increased while the number of reagent samples for the studied viruses has decreased over the years. A total of 27,358 (3.5%) individuals tested positive for at least one type of virus studied. Some of them had more than one simultaneous infection increasing the total to 28,067 infections, and the overall prevalence of infections to 3.6%. The overall positivity rates for HIV, HTLV, HCV, and HBV were 0.4% (2929), 0.3% (2162), 0.4% (3146), and 2.6% (19,830), respectively.

Overall, male donors had comparatively higher frequencies of HIV (64.4% versus 35.6%), HTLV (50.7% versus 49.3%), HCV (63.3% versus 36.7%) and HBV (65.2% versus 34.8%) infection than female ones. There was also a higher rate of infections among donors aged 25-44 years (least in the HIV group in which it was more prevalent infections among donors aged 16-24 years), who had 8–11 years of schooling, living without a partner (with the exception of HBV group) and among other races. For place of residence, it was observed that number of infections was similar between groups in HIV, less prevalent in people that doesn’t live in Bahia for HTLV and HCV infections, and higher in donors who lived in inland cities for HBV infection (Table 4 and Table 5).

Among the donors with a positive -test, 97.5% tested positive for only one virus, 2.4% for 2 virus and 0.1% had triple infection. The average age of the coinfected was 39 years, and males predominated (67.2% of those infected by two viruses and 75.0% of those infected by 3 viruses). Co-infection was more common among individuals with 8–11 years of schooling (49.1%), black/racially mixed (59.9%) and among those without a stable partner (66.2%). The micro-region of Salvador had, proportionally, the highest prevalence of coinfections (58.3%), followed by the micro-regions of Porto Seguro (9.6%), Feira de Santana (4.7%), and Valença (4.6%).

The most common co-infections were HIV/HBV and HCV/HBV, both with the same frequency (30.4%). The frequency of co-infections was comparatively higher in 2009 (13.7%) and lower in 2014 (1.5%).

### 3.3. Georeferencing

Among the 417 municipalities in the state of Bahia, 372 (89.2%) had blood donors with positive results. The distribution by geographic location, based on the donor’s city of residence, showed that the metropolitan region of Salvador (among the mesoregions), and the city of Salvador (among municipalities), had the highest number of cases of infection by the four viruses studied. Feira de Santana (the second largest city in the state), located in the central north region of Bahia, had the second highest rate for HIV, HTLV and HCV; and Eunápolis, located in the south of Bahia, had the second highest rate for HBV. The region of the São Francisco Valley had the lowest rates of infection of the viruses studied, and Juazeiro was the city in this region that had the highest rates (Table 6 and Figure 1, Figure 2, Figure 3 and Figure 4).

Figure 1, Figure 2, Figure 3 and Figure 4 represents the geographic distribution of the HIV infected blood donors identified in Bahia by municipality (Figure 1), HTLV (Figure 2), HCV (Figure 3) and HBV (Figure 4) in the last 10 years in the state of Bahia, by municipality of residence of the donors. Regionally, donors from the Metropolitan Region of Salvador had the highest infection rate (54.5%) and donors from São Francisco Valley had the lowest infection rate (2.8%).

A total of 7224 donors (1.5%) were identified from 541 cities outside the State of Bahia that were not included in the cartographic representations because they reside outside the limits of the state, corresponding to 8832 donations (1.1%).

### 3.4. Density-Type Incidence (DTI)

Of 470,964 donors, 135,490 made more than one donation during the study period with 2064 incident viral infections (456 HIV, 285 HTLV, 558 HCV and 765 HBV). Overall, the incidence density showed a decreasing trend over time for each marker (Table 7), with slightly increasing in the last biennium. The mean time between the 1st donation with negative serology and the 1st positive serological result was 993 days (range 7–3486) considering all viruses together and 897 days (ranging from 61–3372) for HIV, 1071 days (ranging from 61–3426) for HTLV, 1101 days (ranging from 62–3486) for HCV and 945 days ranging from (ranging from 7–3482) for HBV. The average number of donations between the negative and the first positive serologies was 3.1 donations (range 2–42) for all viruses and 3.1 donations (range 2–42) for HIV, 3 donations (range 2–12) for HTLV, 3.2 donations (range 2–18) for HCV and 3 donations (range 2–23) for HBV.

### 3.5. NAT Yield

After the NAT implementation, a seronegative donation for HIV and positive in the NAT was detected (1 in 219,611 donations—0.46/100,000 donations) and 3 donations seronegative for HBV and positive in the NAT (3 in 166,333 donations—1.8/100,000 donations). There was no donor on serological window for HCV after NAT implementation.

## 4. Discussion

In the present study, we evaluated serological positivity for HIV, HTLV, HCV and HBV, in the state of Bahia, over a period of almost 10 years. We detected an overall prevalence of 3.6% for at least one of the routinely screened viruses in blood banks, in Bahia, Brazil. We also detected a decreasing trend in the incidence of such infections overtime. The use of NAT prevented at least 4 infections in this period. The main risks associated with presence of one of the screened infections were male sex, lower education, and being single/divorced. The highest prevalence rates were detected in the mostly populated areas of the state.

Bahia is the largest state in the Northeast region, the 4th largest state in Brazil and was responsible for 5.8% of blood collections in the country in 2016 [8]. Regarding the profile of blood donors, Bahia has a predominance of male donors aged between 25–44 years. This donor profile it is similar that of a private hemotherapy service in our state [15], as well as in other states in Brazil such as Santa Catarina and Pará [16,17] and in other countries such as Iran [18], Italy [19], China [20] and Japan [21]. Bahia has a large percentage of resident people who are black and mixed race [22], which is also reflected in the greater representation of this race found among blood donors (60.1%). 

The number of positive donors for each virus studied was higher in males, to some extent expected, since most donors were male and this is also in agreement with other studies [15,16,23]. Additionally, women seek health services more frequently, which can lead to earlier diagnoses, especially of infections transmitted by transfusion (in prenatal screening, for example), reducing the probability of positive donation of this population. Other significant factors related to positive viral serology were fewer years of schooling and single and single/divorced marital status (without a partner). This draws attention to the need of having specific criteria to select individuals at lower risk for infections in the general population. This measure avoids the inclusion of individuals in the donor cohort with an increase in the number of behaviors at greater risk of acquiring infectious diseases that can be transmitted through transfusion. These measures are even more important as the majority of those infected belong to the most prevalent extracts of donors. Other races (Table 2), including indigens and oriental subjects, showed a higher risk for TTI in this study, than whites or blacks. One potential explanation would be this group is composed by minority populations, which can have poor access to regular health services, leading them to seek blood banks as a substitute way to get serological results and further access to regular healthcare.

With the evolution of the Brazilian hemotherapy legislation, which expanded the age range for donation since 2011 (currently between 16–69 years of age), as well as the constant campaigns in the media and the expansion of Hemoba’s blood collection network, there was a progressive increase in blood donations during the study period. In parallel, there was, in general, a reduction in serological positivity, which may demonstrate an improvement in the selection criteria and professionals’ accuracy who perform clinical screening. Another factor that may explain this risk reduction may be the current existence of serological screening centers available in the State, reducing the number of individuals seeking blood donation to perform tests for the purpose of serological testing.

The global serological positivity found in the present study was 3.5%, a value very close to that described in countries such as Eritrea (3.7%) [24] and close to that found in another service in Bahia (3.13%) [15], but higher than the found in countries such as the United States (0.12%) [25]; Netherlands (0.11%) [23] and Iran (0.254%) [18]. Among the viruses studied, the highest rates of infections were related to HBV (2.6%), mainly due to the positivity of Anti-HBc, a serological marker that remains positive indefinitely and may be the only positive serologic marker in the screening after donation. About HBV, this profile is similar to South Africa [26]. In descending order, there was greater positivity for HCV (0.4%), HIV (0.4%) and HTLV (0.3%), which is consistent with other studies realized in Brazil [26,27,28] and other countries [23,24].

Data analysis although showing a progressive global decrease in viral infections of blood donors, shows an increase in the years 2010 and 2011 of HBV infection, followed by an also progressive decrease. According to information from the Hemoba, in this period, blood collection was expanded in external areas (in places with large circulation of people), using mobile blood collection units, which generated greater access to donation and, consequently, more people were able to donate (sporadic donors), and this may have increased positivity levels. Still in relation to HBV, from 2011 onwards, there was a steady increase in the age group for vaccination until reaching adults up to 59 years old, as part of the national vaccination program (previously restricted to a more specific public, such as health care professionals and people younger than 19 years). That change may be responsible, at least in part, for the decreasing rates of this infection presented over the 10-year period of the study. As shown in Table 5, the prevalence of HBV among blood donors is inversely proportional to the age range, being higher among older donors.

Prevalence rates of HIV, HTLV and HCV infection, were stable across the study period, with values < 1.0%, except in 2008, when HIV infection were above this value (1.4%). As most of Hemoba’s blood donors are repeat donors (53.2%), this behavior may suggest that these donors are concerned with the transfusion safety of recipients of donated blood. The values show low infection rates, which may also be related to the voluntary, altruistic, and unpaid nature of blood donation in Brazil, encouraging self-care and distancing from risk factors for blood-borne infections.

Regarding HTLV, a previous study [29] showed a seroprevalence of 1.35% among blood donors in Salvador, being lower in the present study, but higher than the seroprevalence found in the city of Ribeirão Preto, in the state of São Paulo (0.1%) [30], demonstrating the heterogeneity of infection in different regions of Brazil.

When evaluating co-infected donors, HIV/HBV and HCV/HBV co-infections occurred more commonly, which may be related to the nature of transmission of these viruses through sexual contact and/or by injecting drug use, affecting mostly male donors and with a higher mean age than found for the donor population (39 years old). This may be related to the fact that HBV and HCV infection have a chronic character, being diagnosed accidentally during blood donations, for example, or late when the infected individual already has complications related to the disease, such as cirrhosis and hepatocellular carcinoma [31,32].

Regarding DTI, the present study showed very high incidence rates compared to other countries such as Spain [33] and Canada [34] and close to the incidence of Lithuania [35]. These reports present a higher number of donations than Brazil, in similar periods. The smaller number of donations in Brazil and a likely higher proportion of high-risk donors could impact on incidence and partly explain the present findings. According to data from the Ministry of Health (MS) of Brazil, only 16 out one thousand inhabitants are blood donors in the country, which corresponds to 1.6–1.8% of the Brazilian population, a picture also observed in Bahia [8]. As the donation culture is still incipient in our country, and the access to public health services sometimes is difficult, at-risk subjects, seeking for serological testing, could donate blood just to get tests done, which could lead to the higher incidence rates we observed. The data also indicate a need to improve public policies for the diagnosis and prevention of these infections.

With the implementation of the nucleic acid test for HBV, 3 donors who were in the immunological window had an infection diagnosed (1.8/100,000 donations), thus avoiding transfusional transmission. Considering that the implementation of the NAT for HBV was the last to be implemented globally and mandatory in the country’s hemotherapy services, therefore with less time of use than the NAT for HIV and HCV, this measure represents an important gain for safety transfusion. In Pará [17], a state in northern Brazil, a study observed that the relative risk of HBV infection was estimated at 0.69/100,000 donations. Similarly, HIV transfusion transmission was prevented in 1 donor in the immunological window (0.46/100,000 donations). In Santa Catarina, a state in southern Brazil, another study [16] observed that the residual risk for HIV infection was 1.01/100,000 people per year.

The georeferenced distribution of viral infections studied here shows the areas with the highest infection’s rates according to the city of residence of donors. The greater HIV positivity in the metropolitan region of Salvador may reflect aspects related to the highest population concentration in the state, but may also reflect behavioral aspects, which are often difficult to identify. For the other viruses, the predominance of infection in donors residing outside the state capital, with areas of peak in certain regions (example: higher rate of HBV infection in the south of Bahia, an area with high tourist traffic) may also reflect local behavioral aspects and for HBV, a low vaccination coverage.

A higher concentration of infections is also observed in the largest cities, which are the hubs of the mesoregions described in the georeferencing. This behavior suggests that urbanization and urban agglomerations can influence the spread of these infections.

The present study was carried out with retrospective data, which may lead to the loss of some data, especially due to changes in hemotherapy legislation that took place over the years of the study. To minimize this, the study covered a long period (almost 10 years), in addition to presenting a sizable number of donations, which represents more than half of the reported donations in the State of Bahia.

In summary, the overall prevalence and incidence of blood-borne infections in Bahia showed a decreasing curve over the period studied, which may reflect an improvement in the criteria for screening donors by the service studied (both the implementation of more sensitive and specific tests, and greater training of screening professionals) and the influence of the voluntary, altruistic, and unpaid nature of blood donation in Brazil. 

The results of this study support specific strategies for public health actions for the different mesoregions of the State. This can make the actions of prevention and control of blood-borne infections studied here more effective.

## Figures and Tables

**Figure 1 viruses-14-02323-f001:**
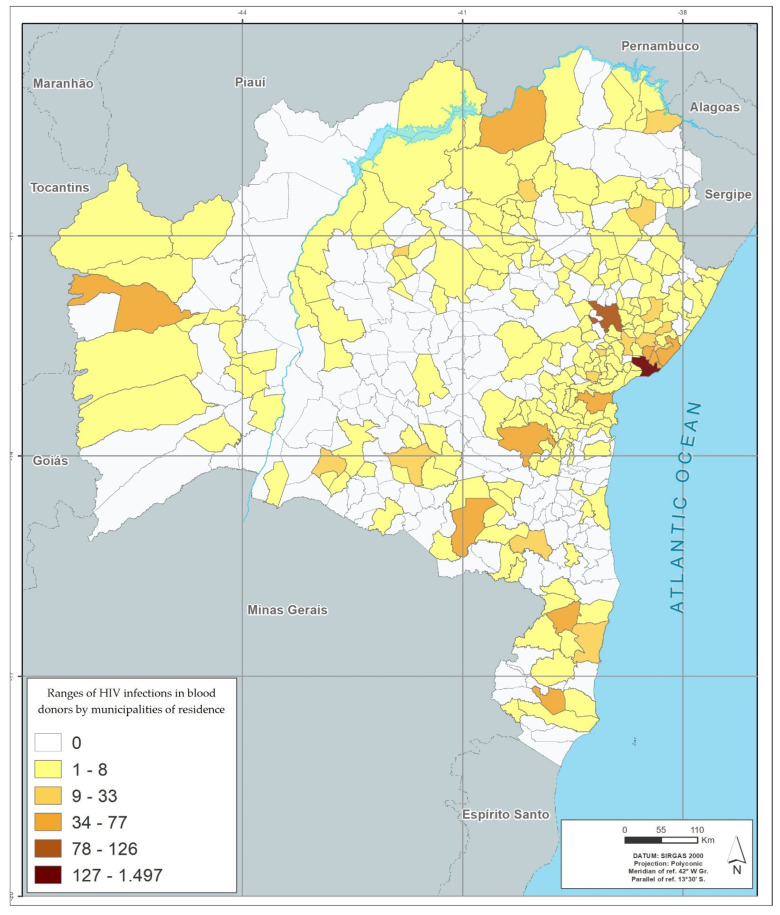
Infection’s rates in absolute numbers for HIV by municipalities of residence in blood donors between 2008–2017.

**Figure 2 viruses-14-02323-f002:**
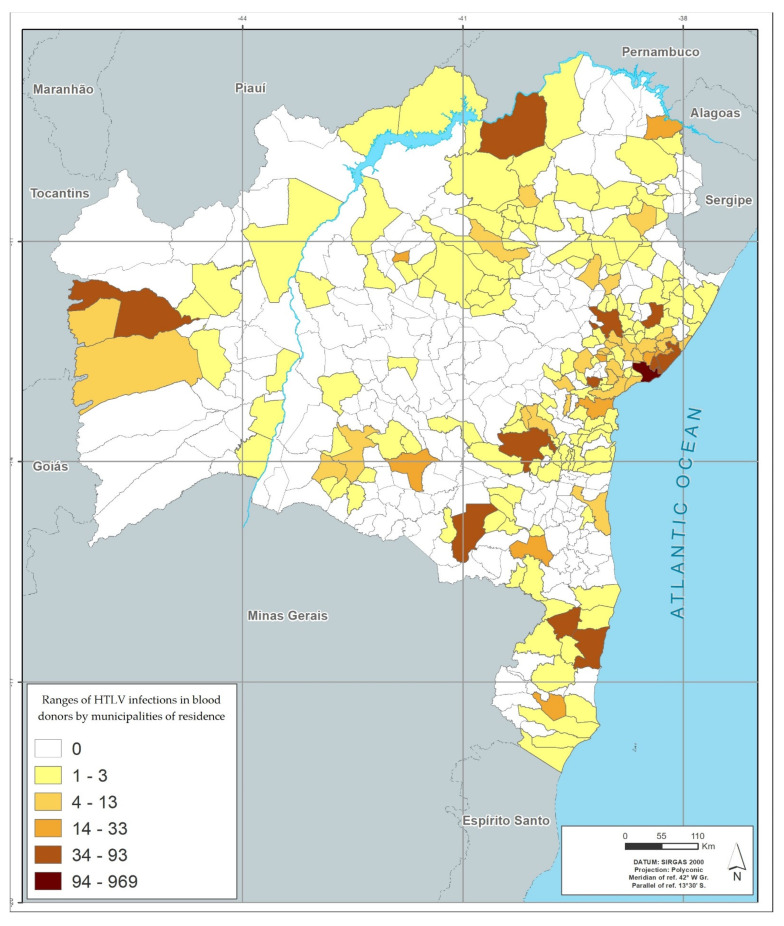
Infection’s rates in absolute numbers for HTLV by municipalities of residence in blood donors between 2008–2017.

**Figure 3 viruses-14-02323-f003:**
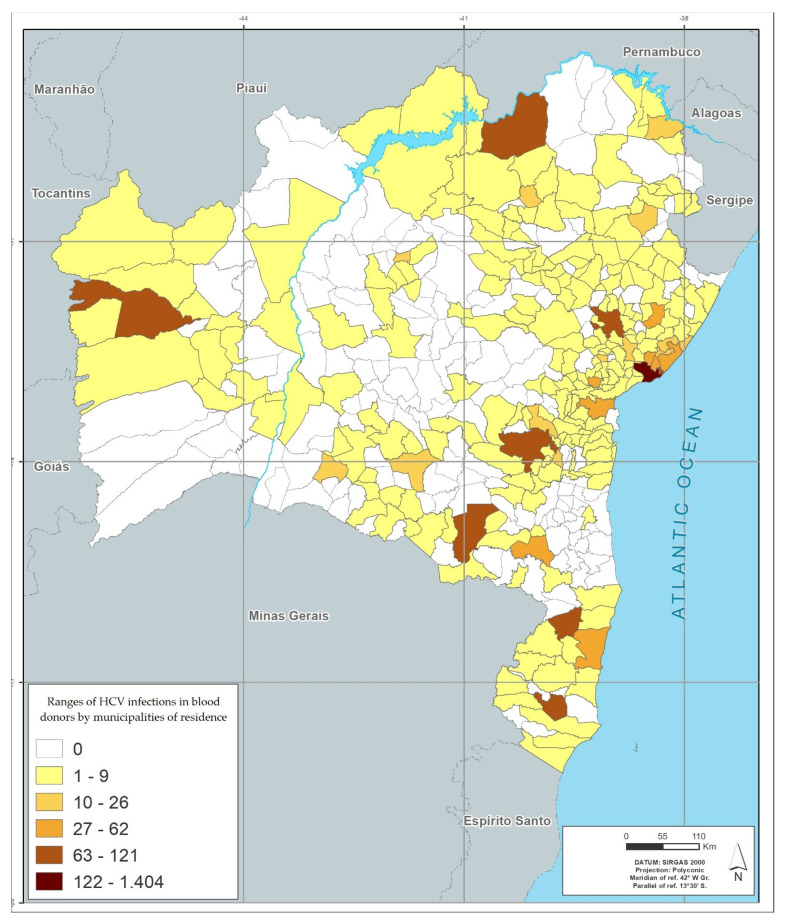
Infection’s rates in absolute numbers for HCV by municipalities of residence in blood donors between 2008–2017.

**Figure 4 viruses-14-02323-f004:**
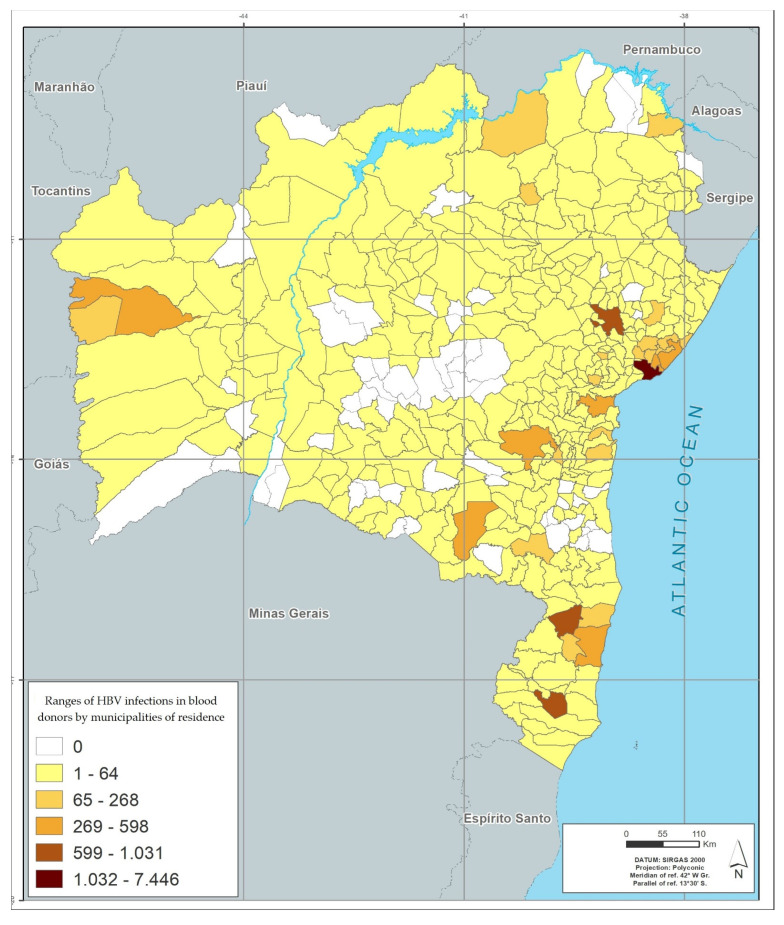
Infection’s rates in absolute numbers for HBV by municipalities of residence in blood donors between 2008–2017.

**Table 1 viruses-14-02323-t001:** Screening tests used at Hemoba from 2008 to 2017.

Methodology and Manufacturer	HIV	HTLV	HCV	HBV	
				HBsAg	Anti-HBc
CMIA/Abbott	2008 to 2017	2015 to 2017	2016 to 2017	2008 to 2017	2011 to 2017
CMIA/Roche	2017	**-**	2017	2017	**-**
EIA/Abbott	**-**	2008 to 2014	**-**	2008 to 2010	**-**
EIA/Biomerieux	**-**	**-**	2008 to 2010	2008 to 2010	2010, 2011, 2014
EIA/Bio-Rad	**-**		2011 to 2016	**-**	2008 to 2013
EIA/Murex	**-**	**-**	2015 to 2016	**-**	**-**
EIA/Ortho	**-**	2010, 2012 to 2015	2016	**-**	**-**
EIA/REM	**-**	**-**	2010 to 2016	**-**	**-**
EIA-I/Abbott	2008 to 2009	**-**	**-**	**-**	**-**
EIA-IC/Abbott	2009 to 2010	**-**	**-**	**-**	**-**
EIA-IC/Murex	2009	**-**	**-**	**-**	**-**
EIA-II/Abbott	2014	**-**	**-**	**-**	**-**
EIA-II/Biomerieux	2008 to 2014	**-**	**-**	**-**	2010
EIA-II/Siemens	2012	**-**	**-**	**-**	**-**
NAT	2013	**-**	2013	2015	

EIA: Enzyme immunoassay; CMIA: Chemiluminescent assay; NAT: Nucleic acid testing. Notes: The immunological window for HCV varied in the study period from 70 (3rd generation test for antibodies detection) to 26 days (4th generation tests, antigen-antibodies detection combination), being reduced to 11 days with NAT. For HBV, the window varies from 25 days (EIA antibodies detection) to 59 days (HBsAg detection), that reduced to 14 days with the use of NAT. For HIV, the window ranged from 23 days (3rd generation Antibodies) detection test to 12–14 days (antigen-antibodies combination detection and NAT). For HTLV, the seroconversion window remains poorly defined, regardless of the serological test used.

**Table 2 viruses-14-02323-t002:** Sociodemographic characteristics of Hemoba seronegative and seropositive blood donors for any viruses from February 2008 to December 2017.

Donor Characteristic	Overall Donations (%)		Negative N (%)		Positive N (%)		Prevalence Rate	*p* Value
Sex								
Female	288,957	37.2	279,028	37.2	9929	36.3	3.44	0.002
Male	488,489	62.8	471,060	62.8	17,429	63.7	3.57	
Age groups								
16–24	168,335	21.7	164,304	21.9	4031	14.7	2.39	<0.0001
25–44	478,153	61.5	461,607	61.5	16,546	60.5	3.46	
45–69	130,958	16.8	124,177	16.6	6781	24.8	5.18	
Race								
White	308,085	39.6	297,522	39.7	10,563	38.6	3.43	<0.0001
Black /Racially mixed	467,280	60.1	450,719	60.1	16,561	60.5	3.54	
Other	2081	0.3	1847	0.2	234	0.9	11.24	
Education								
Up to 1st degree	174,158	22.4	164,504	21.9	9654	35.3	5.54	<0.0001
8–11 schooling years	434,133	55.8	419,702	56.0	14,431	52.7	3.32	
More than 11 schooling years	169,155	21.8	165,882	22.1	3273	12.0	1.93	
Marital status N								
Living without a partner	489,227	62.9	471,851	62.9	17,376	63.5	3.55	0.04
Living with a partner	288,219	37.1	278,237	37.1	9982	36.5	3.46	
Type of blood donor N (%)								
First-time donor	363,828	46.8	342,491	45.7	21,337	78.0	5.86	<0.0001
Repeated donor	413,618	53.2	407,597	54.3	6021	22.0	1.46	
Residential status								
Salvador (capital)	347,047	44.7	336,092	44.8	10,955	40.0	3.16	
Other cities	421,612	54.2	405,437	54.1	16,175	59.1	3.84	<0.0001
Does not live in the state of Bahia	8787	1.1	8559	1.1	228	0.8	2.59	

**Table 3 viruses-14-02323-t003:** HIV, HTLV, HCV, and HBV prevalence rates of Hemoba blood donors in the period from February/2008 to December/2017.

Year	Donations (%)	HIV+ N (%)	HTLV+ N (%)	HCV+ N (%)	HBV+ N (%)	HBsAg+ N (%)	Anti-HBc+ N (%)	Total+ N (%)
2008	36,280 (4.7)	490 (1.4)	108 (0.3)	186 (0.5)	1482 (4.1)	234 (0.6)	1338 (3.7)	2266 (6.2)
2009	45,098 (5.8)	403 (0.9)	131(0.3)	193 (0.4)	1751 (3.9)	202 (0.4)	1657 (3.7)	2478 (5.5)
2010	55,428 (7.1)	251 (0.5)	249 (0.4)	279 (0.5)	2168 (3.9)	224 (0.4)	2058 (3.7)	2947 (5.3)
2011	70,170 (9.0)	269 (0.4)	219 (0.3)	436 (0.6)	2408 (3.4)	183 (0.3)	2359 (3.4)	3332 (4.7)
2012	78,011 (10.0)	261 (0.3)	223 (0.3)	360 (0.5)	2345 (3.0)	163 (0.2)	2314 (3.0)	3189 (4.1)
2013	82,301 (10.6)	353 (0.4)	153 (0.2)	362 (0.4)	2004 (2.4)	144 (0.2)	1964 (2.4)	2872 (3.5)
2014	89,280 (11.5)	282 (0.3)	128 (0.1)	352 (0.4)	2033 (2.3)	203 (0.2)	1941 (2.2)	2795 (3.1)
2015	99,861 (12.8)	178 (0.2)	347 (0.3)	305 (0.3)	1913 (1.9)	187 (0.2)	1815 (1.8)	2743 (2.7)
2016	108,578 (14.0)	156 (0.1)	265 (0.2)	362 (0.3)	1937 (1.8)	192 (0.2)	1840 (1.7)	2720 (2.5)
2017	112,439 (14.5)	286 (0.3)	339 (0.3)	311 (0.3)	1789 (1.6)	149 (0.1)	1734 (1.5)	2725 (2.4)
TOTAL	777,446 (100)	2929 (0.4)	2162 (0.3)	3146 (0.4)	19,830 (2.6)	1881 (0.2)	19,020 (2.4)	28,067 (3.6) *

* This number reflects the total of infections, which includes simultaneous infections in the same donor.

**Table 4 viruses-14-02323-t004:** Bivariate analysis of demographic characteristics associated with HIV and HTLV’s infections among Hemoba blood donors from February 2008 to December 2017.

Characteristics	HIV			HTLV		
	Negative	Positive	Odds Ratio (OR) (95% CI)	Negative	Positive	OR (95% CI)
Number of donations						
First donation	361,907	1921	1	362,266	1562	1
Repeat donation	412,61	1008	0.46 (0.43–0.50)	413,018	600	0.34 (0.31–0.37)
Gender						
Female	287,915	1042	1	287,892	1065	1
Male	486,602	1887	1.07 (0.99–1.16)	487,392	1097	0.61 (0.56–0.66)
Age groups						
16–24	167,620	715	1	167,935	400	1
25–44	476,332	1821	0.90 (0.82–0.98)	476,932	1221	1.07 (0.96–1.20)
45–69	130,565	393	0.71 (0.62–0.80)	130,417	541	1.74 (1.53–1.98)
Education						
Up to 1st degree	173,349	809	1	173,476	682	1
8–11 schooling years	432,51	1623	0.80 (0.74–0.88)	432,968	1165	0.68 (0.62–0.75)
More than 11 schooling years	168,658	497	0.63 (0.56–0.71)	168,84	315	0.47 (0.42–0.54)
Marital status						
Living without a partner	487,151	2076	1	487,776	1451	1
Living with a partner	287,366	853	0.70 (0.64–0.75)	287,508	711	0.83 (0.76–0.91)
Race						
White	306,942	1143	1	307,213	872	1
Black/ Racially mixed	465,524	1756	1.01 (0.94–1.09)	466,005	1275	0.96 (0.88–1.05)
Other	2051	30	3.93 (2.73–5.66)	2066	15	2.56 (1.53–4.27)
Residential status						
Salvador (capital)	345,549	1498	1	346,078	969	1
Other cities	420,22	1392	0.76 (0.71–0.82)	420,428	1184	1.01 (0.92–1.10)
Doesn’t live in Bahia	8748	39	1.03 (0.75–1.41)	8778	9	0.37 (0.19–0.71)

**Table 5 viruses-14-02323-t005:** Bivariate analysis of demographic characteristics associated with HCV and HBV’s infections among Hemoba blood donors from February 2008 to December 2017.

Characteristics	HCV			HBV		
	Negative	Positive	OR (95% CI)	Negative	Positive	OR (95% CI)
Number of donations						
First donation	361,764	2064	1	347,439	16,389	1
Repeat donation	412,536	1082	0.46 (0.43–0.49)	410,177	3441	0.18 (0.17–0.18)
Gender						
Female	287,802	1155	1	282,061	6896	1
Male	486,498	1991	1.02 (0.95–1.10)	475,555	12,934	1.11 (1.08–1.15)
Age groups						
16–24	167,739	596	1	165,934	2401	1
25–44	476,282	1871	1.11(1.01–1.21)	466,151	12,002	1.78 (1.70–1.86)
45–69	130,279	679	1.47 (1.31–1.64)	125,531	5427	2.99 (2.85–3.14)
Education						
Up to 1st degree	173,251	907	1	166,612	7546	1
8–11 schooling years	432,398	1735	0.77 (0.71–0.83)	423,878	10,255	0.53 (0.52–0.55)
More than 11 schooling years	168,651	504	0.57 (0.51–0.64)	167,126	2029	0.27 (0.26–0.28)
Marital status						
Living without a partner	487,196	2031	1	476,942	12,285	1
Living with a partner	287,104	1115	0.93 (0.87–1.00)	280,674	7545	1.04 (1.01–1.07)
Race						
White	306,853	1232	1	300,491	7594	1
Black/Racially mixed	465,392	1888	1.01 (0.94–1.09)	455,213	12,067	1.05 (1.02–1.08)
Other	2055	26	3.15 (2.13–4.66)	1912	169	3.50 (2.98–4.10)
Residential status						
Salvador (capital)	345,643	1404	1	339,601	7446	1
Other cities	419,896	1716	1.01 (0.94–1.09)	409,386	12,226	1.36 (1.32–1.40)
Doesn’t live in Bahia	8761	26	0.73 (0.50–1.08)	8629	158	0.84 (0.71–0.98)

**Table 6 viruses-14-02323-t006:** Prevalence of HIV, HTLV, HCV and HBV infections by mesoregions of residence in the State of Bahia, from February 2008 to December 2017.

Bahia Mesoregions	Total	HIV	%	HTLV	%	HCV	%	HBV	%
Metropolitan Region of Salvador	441,826	1882	0.4	1301	0.3	1864	0.4	10,184	2.3
South Bahia	79,374	265	0.3	207	0.3	305	0.4	4220	5.3
Central-South Bahia	74,991	214	0.3	215	0.3	318	0.4	1500	2.0
Central-North Bahia	71,528	235	0.3	192	0.3	260	0.4	1699	2.4
Far-West Bahia	36,306	98	0.3	81	0.2	108	0.3	786	2.2
Northeast Bahia	33,449	108	0.3	93	0.3	148	0.4	780	2.3
São Francisco Valley of Bahia	31,140	88	0.3	64	0.2	117	0.4	503	1.6
Total	768,614	2890	0.4	2153	0.3	3120	0.4	19,672	2.6

**Table 7 viruses-14-02323-t007:** Incidence Density rates (95% CI) of HIV, HTLV, HCV and HBV infection in Hemoba blood service donors, 2008–2017.

Year	Person-Years	HIV +	Incidence per 100,000 (95% CI)	HTLV +	Incidence per 100,000 (95% CI)	HCV +	Incidence per 100,000 (95% CI)	HBV +	Incidence per 100,000 (95% CI)
		(N)		(N)		(N)		(N)	
2008–2009	28,290	43	152.0 (106.6–197.4)	3	10.6 (0–22.6)	16	56.6 (28.9–84.3)	30	106.0 (68–144)
2010–2011	100,240	57	56.9 (42.1–71.6)	39	38.9 (26.7–51.1)	94	93.8 (74.8–112.7)	199	198.5 (170.9–226.1)
2012–2013	188,743	133	70.5 (58.5–82.4)	32	16.9 (11.1–22.8)	84	44.5 (35–54)	155	82.1 (69.2–95.1)
2014–2015	269,924	95	35.2 (28.1–42.3)	100	37.1 (30–44.3)	119	44.1 (36.2–52)	175	64.8 (55.2–74.4)
2016–2017	252,445	128	50.7 (41.9–59.5)	111	44 (35.8–52.1)	245	97.1 (84.9–109.2)	206	81.6 (70.5–92.7)
TOTAL	839,642	456	54.3 (49.3–59.3)	285	33.9 (30.0–37.9)	558	66.5 (60.9–72.0)	765	91.1 (84.6–95.7)

## Data Availability

All data are available under request to the corresponding author.

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
