# Peer review of "HIV, HTLV, and Hepatitis B and C Infection in Blood Donors in Bahia, Brazil from 2008 to 2017"

_viruses, 2022, doi:10.3390/v14112323_

Round 1

Reviewer 1 Report

The manuscript deals with the time trend in the prevalence of viral infections (HIV, HTLV, hepatitis B, and hepatitis C) among blood donors in a Brazilian state over the 2008-2017 period,  during which NAT screening for HIV and hepatitis infections was introduced.  Although the theme is undoubtedly relevant, some major methodological improvements are needed before reaching publication quality.

Major issues – Key problem is focusing on the period prevalence and annual cumulative incidence (ACI) instead of the density-type incidence (DTI). The latter is a standard measure in the field of transfusion-transmitted infections (TTI) because its multiplication with the probability of infection during the immunological window period – proportional to its duration – expresses the residual TTI risk after blood screening. It has been widely used in the TTI literature and would therefore allow better comparability with other studies in the field.

Annual prevalence may be a poor approximation to DTI when the average time between the last test-negative and first test-positive result is 993 days (2.72 years) as reported in this study. The standard incidence-window method uses the midpoint between the dates corresponding to these test results to estimate the likely date of viral infection for repeat donors, which in turn allows the calculation of person-time at risk for all of these donors. Period prevalence can be used to compare the first-time and repeat donors.

Another serious shortfall is the absence of precision estimates, such as a 95% confidence interval (CI), typically based on the Poisson distribution. The CI was mentioned in the title of Table 6 but omitted from the table.

Specific issues

Abstract – “positivity rates” is a vague expression for the ACI. Calculate DTI and report it here.

Ad 2.1. - This seems to be an open cohort study. Substitute ACI with DTI as suggested above. Care must be taken to truncate person-time at the start and the end of the period analyzed. Instead of an annual time trend, perhaps a biannual trend would better smooth out large variations for rare events.

Table 1 should add the information on immunological window duration for each test, including the NAT.

Briefly describe the Brazilian Ministry of Health's definition of test-positive results in blood screening for each infection under study as the reference [4] is not widely known to the journal readers.

Ad 2.4.1. – You say “Person-years were calculated by adding the follow-up time of all donors, in which a donor's follow-up time was the time between the first and last donation in any interval of one year". As the mean interdonation period for converting donors was 2.72 years, individual person-time at risk should be calculated for all donors within each year of the period analyzed, with different formulas for repeat and first-time donors. Please provide these formulas in the revised manuscript. The above citation contradicts the sentence that follows it "The number of donors who seroconverted were divided by the number of donations in that year and the rate was expressed per 100,000 person-years” because it amounts to ACI, whereas the sentence that precedes it suggests DTI. Again, the latter should be used as the preferred incidence measure, and the details of its calculation should be crystal clear.

3.1. In the first sentence, you say that up to 102 donations were provided by a single donor over ten years. This donation frequency is not legally permitted, so I suspect these were the dates of repeated blood samples taken to elucidate an infection status and not to make a blood donation. Special care must be given to calculate the correct person-time for such cases.

Table 2 – Add a person-time column and 95% CI for the prevalence estimates. In discussion, comment high risk for the "other" race.

Table 3 - Present DTI instead of ACI. Did you have the means to distinguish between Anti-HBc due to HBV infection versus HBV vaccination?

Table 4 – Move this table in the supplement if you wish but substitute “%” with prevalence ratio and its 95% CI. The percentages presented in the table and cited throughout the text are misleading! Provide DTI with 95% CI instead in the paper.

3.3. Figure 1 legend is unreadable. Substitute “in the last 10 years” with “2008-2017” on line 27 below the figure. The percentages on lines 29-30 are misleading; use risk ratio instead.

3.4. Despite being titled "Incidence", this ten-line paragraph provides only a reference to Table 6 with ACI and the mean time between the first positive and last negative test result for each infection analyzed. Provide DTI and corresponding 95% CI.

4. Discussion – The studies from Eritrea, the Netherlands, and the USA, cited in the present study, are far from being representative of the TTI trend and NAT yield. The results from other Brazilian states have been published and deserve comment in the light of the present study results, as do the findings from other developing countries (South Africa, China) and developed countries in Europe, plus Canada and Australia.

On lines 96-98, it is not clear what was meant by “… the highest rates of infections were related to HBV (2.6%), mainly due to the positivity of Anti-HBc, a serological marker that is detected earlier and remains positive indefinitely". Earlier than what? HBsAg spikes earlier than Anti-HBc, so please clarify the phrase.

Comment if the age of HBV-infected donors is compatible with the hypothesis put forward on lines 106-110.

Author Response

We thank reviewers for their time and effort that contributed to increase the quality of our manuscript. Please, below a point-by-point response to reviewers comments.

Comments and Suggestions for Authors

The manuscript deals with the time trend in the prevalence of viral infections (HIV, HTLV, hepatitis B, and hepatitis C) among blood donors in a Brazilian state over the 2008-2017 period,  during which NAT screening for HIV and hepatitis infections was introduced.  Although the theme is undoubtedly relevant, some major methodological improvements are needed before reaching publication quality.

  1. Major issues – Key problem is focusing on the period prevalence and annual cumulative incidence (ACI) instead of the density-type incidence (DTI). The latter is a standard measure in the field of transfusion-transmitted infections (TTI) because its multiplication with the probability of infection during the immunological window period – proportional to its duration – expresses the residual TTI risk after blood screening. It has been widely used in the TTI literature and would therefore allow better comparability with other studies in the field.

R- We changed the data presentation, replacing prevalence by DTI, as suggested by the reviewer

  1. Annual prevalence may be a poor approximation to DTI when the average time between the last test-negative and first test-positive result is 993 days (2.72 years) as reported in this study. The standard incidence-window method uses the midpoint between the dates corresponding to these test results to estimate the likely date of viral infection for repeat donors, which in turn allows the calculation of person-time at risk for all of these donors. Period prevalence can be used to compare the first-time and repeat donors.

R- We believe that prevalence rates are important to know the associations with demographic information, as shown in tables 4 and 5, which already include the  data for first and repeat donations, providing useful information to understand the overall Picture in the study´s period.

  1. Another serious shortfall is the absence of precision estimates, such as a 95% confidence interval (CI), typically based on the Poisson distribution. The CI was mentioned in the title of Table 6 but omitted from the table.

 R- We agree with reviewer and 95% CI were included in tables, when appropriate.

 Specific issues

  1. Abstract – “positivity rates” is a vague expression for the ACI. Calculate DTI and report it here.

R- DTI was calculated and added to abstract, as suggested

  1. Ad 2.1. - This seems to be an open cohort study. Substitute ACI with DTI as suggested above. Care must be taken to truncate person-time at the start and the end of the period analyzed. Instead of an annual time trend, perhaps a biannual trend would better smooth out large variations for rare events.

R- It was done!

  1. Table 1 should add the information on immunological window duration for each test, including the NAT.

R- A table footnote was added with the immunological window information. Adding the info directly to the table would make it too busy and hard to read.

  1. Briefly describe the Brazilian Ministry of Health's definition of test-positive results in blood screening for each infection under study as the reference [4] is not widely known to the journal readers.

R-Briefly, the blood samples obtained from each donation are tested as defined by the Ministry of Health. Prior to NAT implementation, the samples were tested in high sensitivity kits (ELISA and/or chemiluminescence) for antigen and/or antibody detection against the target viruses, 2 tests were performed for HIV and HBV and 1 test for HCV and HTLV. In case of a positive result, the tests were repeated in duplicate in the same sample and an additional test, using a different methodology should be carried out for confirmation. After the implementation of NAT for HIV and HBV, one of the initial tests was replaced by NAT and, for HCV, NAT was added to the serological test. The testing for HTLV remained the same, considering that there is still no obligation of NAT for HTLV in Brazil.

  1. Ad 2.4.1. – You say “Person-years were calculated by adding the follow-up time of all donors, in which a donor's follow-up time was the time between the first and last donation in any interval of one year". As the mean interdonation period for converting donors was 2.72 years, individual person-time at risk should be calculated for all donors within each year of the period analyzed, with different formulas for repeat and first-time donors. Please provide these formulas in the revised manuscript. The above citation contradicts the sentence that follows it "The number of donors who seroconverted were divided by the number of donations in that year and the rate was expressed per 100,000 person-years” because it amounts to ACI, whereas the sentence that precedes it suggests DTI. Again, the latter should be used as the preferred incidence measure, and the details of its calculation should be crystal clear.

R- It was corrected, as suggested

  1. In the first sentence, you say that up to 102 donations were provided by a single donor over ten years. This donation frequency is not legally permitted, so I suspect these were the dates of repeated blood samples taken to elucidate an infection status and not to make a blood donation. Special care must be given to calculate the correct person-time for such cases.

R- The donor in question (who provided 102 donations) is a male donor, highly loyal to the institution (the data was confirmed by the institution, without identification of the donor to the researchers), who made regular donations of whole blood and by apheresis. In Brazil, men can donate whole blood every 60 days, with a maximum of 4 donations in 12 months. However, platelets donation (by aphresis) can be provided up to 12- 24 donations in 12 months  (depending the amount of platelets collected),. As the donor alternated donations of whole blood and by platelets (by apheresis) and regularly donated during the 10 years of the study, the data is perfectly possible and correct.

  1. Table 2 – Add a person-time column and 95% CI for the prevalence estimates. In discussion, comment high risk for the "other" race.

R- It was added, as well as a comment on race

  1. Table 3 - Present DTI instead of ACI. Did you have the means to distinguish between Anti-HBc due to HBV infection versus HBV vaccination?

             R- I think there was a mistake in the comment, as vaccination provide antibodies against HBV surface antigen (Anti HBs), but not against HBc (Anti HBc)

  1. Table 4 – Move this table in the supplement if you wish but substitute “%” with prevalence ratio and its 95% CI. The percentages presented in the table and cited throughout the text are misleading! Provide DTI with 95% CI instead in the paper.

R-. We changed it. However, because this made the table too busy and hard to read, we had to split it in two tables (data on two viruses each).

  1. Figure 1 legend is unreadable. Substitute “in the last 10 years” with “2008-2017” on line 27 below the figure. The percentages on lines 29-30 are misleading; use risk ratio instead.

R- Done!

  1. Despite being titled "Incidence", this ten-line paragraph provides only a reference to Table 6 with ACI and the mean time between the first positive and last negative test result for each infection analyzed. Provide DTI and corresponding 95% CI.

        R- It was modified accordingly

  1. Discussion – The studies from Eritrea, the Netherlands, and the USA, cited in the present study, are far from being representative of the TTI trend and NAT yield. The results from other Brazilian states have been published and deserve comment in the light of the present study results, as do the findings from other developing countries (South Africa, China) and developed countries in Europe, plus Canada and Australia.

R- New references were added, as suggested, and discussion was modified accordingly.

  1. On lines 96-98, it is not clear what was meant by “… the highest rates of infections were related to HBV (2.6%), mainly due to the positivity of Anti-HBc, a serological marker that is detected earlier and remains positive indefinitely". Earlier than what? HBsAg spikes earlier than Anti-HBc, so please clarify the phrase.

            R-Thanks for the comment. We modified the sentence to make it clearer

  1. Comment if the age of HBV-infected donors is compatible with the hypothesis put forward on lines 106-110.

R- a comment on this issue was added to text, showing that prevalence of Anti HBV antibodies increases with age

Reviewer 2 Report

In this retrospective study, the authors evaluated HIV, HTLV, HCV, and HBV infection rates in the blood of the Hematology and Hemotherapy Foundation of Bahia (Hemoba), Brazil, through serological and NAT results and the characteristics of donors

In the present manuscript, they show a decreasing trend over the years studied, with a predominance of HBV infection.

In general, the studies are OK and the manuscript could be improved better.

My comments are shown below, the author may either address these comments or add the limitation in the discussion section.

1.      In the text, the font format and size should be consistent, not like Material and Methods 2.1 study design and population, or references sections.

2.      Could the authors change Table3 into figures to better show the results?

3.      In line 54, there show be spaces between 4 and Discussion.

4.      Again, the font format and size are not consistent, and should be improved.

Author Response

In this retrospective study, the authors evaluated HIV, HTLV, HCV, and HBV infection rates in the blood of the Hematology and Hemotherapy Foundation of Bahia (Hemoba), Brazil, through serological and NAT results and the characteristics of donors

In the present manuscript, they show a decreasing trend over the years studied, with a predominance of HBV infection.

In general, the studies are OK and the manuscript could be improved better.

My comments are shown below, the author may either address these comments or add the limitation in the discussion section.

  1. In the text, the font format and size should be consistent, not like Material and Methods 2.1 study design and population, or references sections.

R- Thanks, it was corrected

  1. Could the authors change Table3 into figures to better show the results?

            R- Thanks for the suggestion. We do prefer to keep data on table and text, as provided in the original version, because the resulting figure would be difficult to understand, due to the difference in magnitude of reported information, which would be hard to display in a single table

  1. In line 54, there show be spaces between 4 and Discussion.

            R- It was corrected

  1. Again, the font format and size are not consistent, and should be improved

           R- Fixed!